# Hidden Unstable Flap Should Be Suspected in Treating Intractable Pain from Medial Meniscus Horizontal Tear

**DOI:** 10.3390/jcm11216245

**Published:** 2022-10-23

**Authors:** Young Mo Kim, Yong Bum Joo, Byung Kuk An, Ju-Ho Song

**Affiliations:** 1Department of Orthopedic Surgery, Chungnam National University Hospital, Chungnam National University College of Medicine, Daejeon 35015, Korea; 2Department of Orthopedic Surgery, Chungnam National University Sejong Hospital, Chungnam National University College of Medicine, Sejong 30099, Korea

**Keywords:** medial meniscus, horizontal tear, arthroscopic meniscectomy

## Abstract

(1) Background: The medial meniscus horizontal tear (MMHT) is known as a lesion that can be treated nonoperatively. However, some patients show persistent pain despite conservative treatments. In arthroscopic surgery for MMHT, surgeons often encounter unexpected unstable flaps, which can explain the intractable pain. This study aimed to determine whether preoperative factors could predict the hidden unstable flaps in MMHT. (2) Materials and Methods: Medical records of 65 patients who underwent arthroscopic partial meniscectomy (APM) for isolated MMHT during 2016–2020 were retrospectively reviewed. APM was indicated when there was no severe chondral degeneration and intractable localized knee pain in the medial compartment did not resolve despite conservative treatments. Unstable flap was confirmed based on arthroscopic images and operation notes. Each of the following preoperative factors were investigated using logistic regression analyses to determine whether they can predict an unstable flap: age, sex, body mass index, lower limb alignment, trauma history, mechanical symptoms, symptom duration, visual analogue scale (VAS), Lysholm score, cartilage wear of the medial compartment, and subchondral bone marrow lesion (BML). (3) Results: Hidden unstable flaps were noted in 45 (69.2%) patients. Based on univariate analyses for each preoperative factor, age, symptom duration, cartilage wear (of the femoral condyle and the tibial plateau), and subchondral BML were included in the multivariate logistic regression analysis. The results showed that symptom duration (*p* = 0.026, odds ratio = 0.99) and high-grade cartilage wear of the medial femoral condyle (*p* = 0.017, odds ratio = 0.06) were negatively associated with unstable flaps. A receiver operating characteristic curve was used to calculate the symptom duration at which the prediction of unstable flaps was maximized, and the cutoff point was 14.0 months. (4) Conclusions: More than two thirds of patients suffering intractable pain from MMHT had hidden unstable flaps. However, APM should not be considered when the symptom duration is more than 14 months or high-grade cartilage wear of the medial femoral condyle is noted.

## 1. Introduction

The paradigm has been shifted from “If it is torn, take it out!” to “Save the meniscus!” in the treatment of meniscal tear [1,2]. Several randomized controlled studies showed that arthroscopic partial meniscectomy (APM) had no clinical advantage over conservative treatments [3,4,5,6,7,8]. In particular, the medial meniscus horizontal tear (MMHT) is regarded as a degenerative lesion that should be addressed in a non-surgical approach [6,9,10]. However, not every patient with MMHT can be treated with non-steroidal anti-inflammatory drugs and muscle strengthening exercises. The randomized trials questioning the efficacy of APM included only a subset of patients who responded to conservative treatments [11]. If MMHT is not accompanied by severe chondral degeneration and pain that is not relieved despite conservative treatments, APM is a remaining option to resolve the intractable pain.

Some previous studies found that APM was more effective in patients with mechanical symptoms than in those without these symptoms [12,13]. The meniscal flap, being lodged between the femoral condyle and the tibia plateau, has been recognized as a lesion inducing mechanical symptoms [14,15,16]. However, the unstable flap is often difficult to discern on MRI scans [17,18] and is not detected during arthroscopy unless it is probed using a hook [19,20]. The hidden flap might be the reason that conservative treatments fail in some cases of MMHT.

Recent studies attempted to identify a subgroup of patients who could show particularly favorable surgical outcomes after APM [21,22]. If surgeons preoperatively distinguish between those who would benefit from APM and those who would not, unnecessary surgery could be avoided and the outcomes of APM will be improved. Moreover, there is a growing interest in the arthroscopic repair of MMHT [23,24]. Predicting unstable flaps is closely related to surgical indications. Thus, this study reviewed patients complaining of intractable pain from MMHT that necessitated APM. The incidence of hidden meniscal flaps was investigated and the preoperative predictive factors for the hidden unstable flaps were identified. It was hypothesized that certain preoperative factors differ according to the unstable meniscal flap, which was confirmed based on the arthroscopic findings.

## 2. Materials and Methods

Medical records of 65 patients who underwent APM for isolated MMHT during 2016–2021 were retrospectively reviewed after approval was obtained from the institutional review board of Chungnam National University Hospital (No. 2022-05-075). Patients with a discernable meniscal flap on preoperative magnetic resonance imaging (MRI) scans were not included in this study. The exclusion criteria also included combined ligament injury, lateral meniscus tear, and previous knee surgery. APM was indicated when intractable localized knee pain in the medial compartment did not resolve despite conservative treatments, such as non-steroidal anti-inflammatory drugs and muscle strengthening exercises. Patients showing severe medial OA on radiographs (Kellgren–Lawrence grade ≥ 3) were not deemed suitable for APM.

All APMs were performed by a single senior surgeon. Arthroscopic findings were documented as photos, videos, and intraoperative notes. To resolve mechanical symptoms such as catching, locking, and pain during squatting, the displaceable meniscal flap and unstable superior or inferior leaves of MMHT were resected, while, at the same time, every effort was made to preserve intact meniscal tissue. No patient underwent subtotal or total meniscectomy. Full range of motion and muscle strengthening exercises were encouraged immediately after surgery.

### 2.1. Study Design

The unstable flap was confirmed on arthroscopic images and operation notes (Figure 1). An unstable flap was defined based on the ISAKOS classification: <3 mm displacement on arthroscopic probing [25]. The patients were divided into a flap group and non-flap group. To identify predictive factors for unstable flaps, the following factors were investigated: age, sex, body mass index (BMI), lower limb alignment, trauma history, mechanical symptoms, symptom duration (from the onset of symptoms to surgery), visual analogue scale (VAS), Lysholm score, cartilage wear of the medial compartment, and subchondral bone marrow lesion (BML). For the alignment, varus and valgus mechanical alignments were defined as positive and negative, respectively. The cartilage wear in the medial femoral condyle and the medial tibial plateau was separately assessed on MRI scans according to the Yulish grading system: grade 0, normal; grade 1, normal cartilage contour with abnormal signal; grade 2, superficial fraying, erosion, or ulceration of <50%; grade 3, partial-thickness defect of >50%; and grade 4, full-thickness defect [26]. High-grade cartilage wear was deemed as that of ≥grade 3 [27]. Subchondral BML was defined as a locus of increased signal in the subchondral area with internal trabecular marrow in it [28]. The presence of subchondral BML was investigated on MRI scans. The evaluation of the unstable flap, cartilage wear, and subchondral BML was independently performed by three orthopedic surgeons, and all disagreements were resolved by discussion.

### 2.2. Statistical Analysis

Predictive factors for unstable flaps were identified using logistic regression analyses. Univariate analyses for each factor were performed first; thereafter, a multivariate regression analysis was conducted to avoid overfitting problems. Between the flap group and the non-flap group, categorical variables were analyzed by Chi-square test when the expected value of the cell was 5 or more in at least 80% of the cells; otherwise, Fisher exact test was used. Continuous variables were analyzed by *t*-test. When continuous variables were found to be significant factors in the multivariate regression analysis, a receiver operating characteristic curve (ROC) was used to obtain the cutoff point at which the sensitivity and the specificity for the unstable flap were maximized. All statistical analyses were performed using R software version 4.1.1 (R foundation for Statistical Computing, Vienna, Austria), with a *p* value < 0.05 considered statistically significant.

## 3. Results

Of 65 patients with a mean age of 52.2 ± 12.6 years (range, 18–75 years), hidden unstable flaps were noted in 45 (69.2%) patients. The mean symptom duration was 13.0 ± 5.8 weeks. The flap group and the non-flap group showed significant differences in age, symptom duration, cartilage wear of the femoral condyle and the tibial plateau, and subchondral BML. The patient characteristics among the groups are summarized in Table 1.

### Logistic Regression Analyses Regarding Unstable Flap

Based on univariate analyses, age, symptom duration, cartilage wear of the femoral condyle and the tibial plateau, and subchondral BML were included in a multivariate logistic regression analysis. The results showed that symptom duration (*p* = 0.026, odds ratio = 0.99) and high-grade cartilage wear of the medial femoral condyle (*p* = 0.017, odds ratio = 0.06) were negatively associated with an unstable flap (Table 2). An ROC curve was used to calculate the symptom duration at which the sensitivity and the specificity for unstable flaps were maximized (Figure 2), and the cutoff point was 14.0 months, with the area under curve being 0.70 (sensitivity, 50.0%; specificity, 83.7%).

## 4. Discussion

The most important findings of the present study were that (1) approximately 70% of patients suffering intractable pain had hidden unstable flaps that would have been lodged and could be a source of pain, and (2) the symptom duration and cartilage wear of the medial femoral condyle had negative correlations with unstable flaps. Pain was not likely from an unstable flap if the symptom duration was more than 14 months or high-grade cartilage wear of the femoral condyle was noted on preoperative MRI scans.

MMHT usually occurs in middle-aged patients as a degenerative lesion without a definite trauma history [29,30]. The tear dividing the meniscus into the upper and the lower leaves is often asymptomatic, and, if pain occurs, the patient responds well to conservative treatments [6,31]. When the stable leaf receiving additional damage develops an unstable flap, APM will be required [9,18]. Recently, arthroscopic repair was also considered as a reliable option in treating MMHT because the benefit of meniscal repair had been documented [23,32]. However, it is not always possible to confirm the unstable flap on MRI scans [17,18]. Identification of the predictive factors for unstable flaps can help surgeons to determine the necessity of APM before several months of conservative treatments fail. In their recent study based on the Knee Arthroscopy Cohort of Southern Denmark, Pihl et al. tried to identify a subgroup of patients who would benefit from APM. Although they investigated a number of preoperative factors, including patient demographics and knee-related symptoms, the predictive performance of the prognostic model was poor [21]. Noorduyn et al. also found that there was no clear subgroup of patients who would benefit from either surgery or physical therapy for a degenerative meniscal tear [22].

This study found two significant factors in the logistic regression analysis regarding unstable flaps: symptom duration and cartilage wear of the femoral condyle. Kim et al. also reported similar results on symptom duration [9]. They found that patients with flap tears had a shorter symptom duration to surgery than those without a flap tear (6 months vs. 22 months). However, they did not find a significant difference in the severity of osteoarthritis on plain radiographs (Kellgren–Lawrence grade) according to flap tear. In the present study, the cartilage status of the medial compartment was assessed based on MRI scans, and high-grade cartilage wear of the femoral condyle had a negative correlation with unstable flaps. Thus, the mechanical symptoms of the non-flap group were likely to result from the complex process of osteoarthritis [33].

It is noteworthy that mechanical symptoms and the severity of pain (VAS) did not differ significantly between the flap group and the non-flap group. Because a displaced flap can lead to mechanical symptoms that require arthroscopic management [14,15,16], mechanical symptoms and a positive McMurray test considerably influence a surgeon’s decision to recommend APM [11]. However, they were not useful in predicting unstable flaps. The severity of pain was not a predictive factor either. Regarding this, van de Graaf et al. performed an interesting investigation based on a survey of 194 orthopedic surgeons [34]. Each surgeon was informed about patient profiles, including pain score, the results of physical examination, and the type of meniscal tear. In predicting the outcome of APM, experienced knee surgeons were not better than other orthopedic surgeons. Clinical symptoms were not reliable criteria for deciding whether to use APM.

Several limitations should be noted. First, patients who had conservative treatments for MMHT were not included in this study. The necessity of APM would have been determined better if these patients had been included. Second, patients who underwent APM for MMHT were relatively rare, which made it difficult to attain a large number of patients in this study. The small sample size would compromise the statistical power. Third, this study failed to identify positive predictive factors for unstable flaps. The two significant factors (symptom duration and cartilage wear) were negatively correlated with unstable flaps. Thus, preoperative factors that support the necessity of APM are not revealed.

## 5. Conclusions

More than two thirds of patients suffering intractable pain from MMHT had hidden unstable flaps. However, APM should not be considered when the symptom duration is more than 14 months or high-grade cartilage wear of the medial femoral condyle is noted.

## Figures and Tables

**Figure 1 jcm-11-06245-f001:**
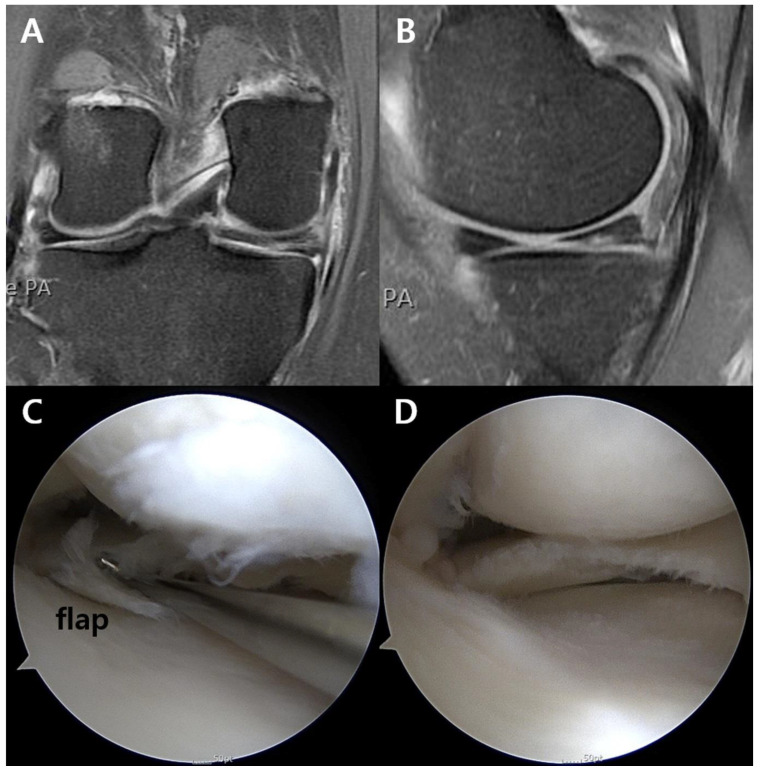
Unstable flap at the posterior horn of the medial meniscus in the right knee. (**A**) Coronal and (**B**) sagittal MRI images did not definitely show the flap, which was (**C**) confirmed and (**D**) debrided during arthroscopy. MRI, magnetic resonance imaging.

**Figure 2 jcm-11-06245-f002:**
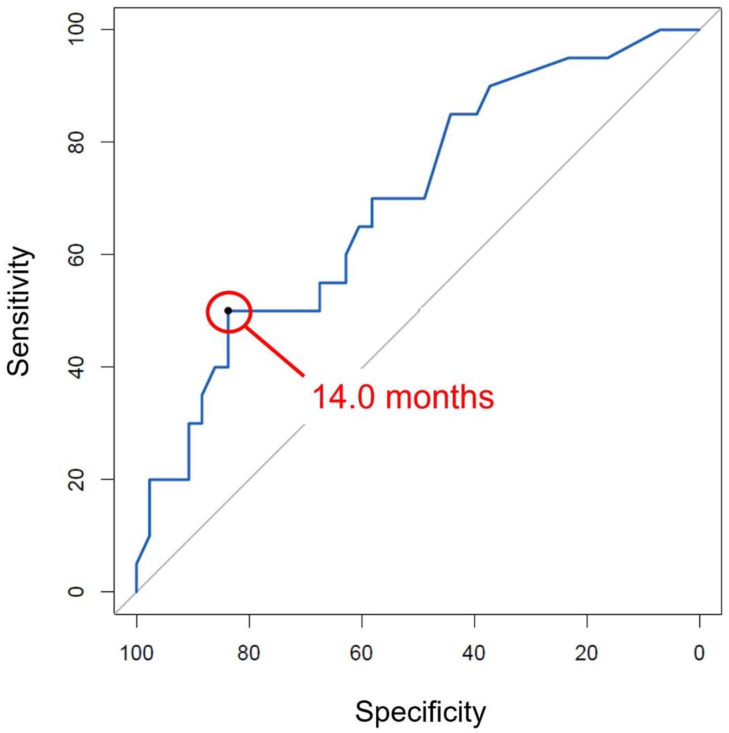
Receiver operating characteristic curve for unstable flap showed that the cutoff point of symptom duration was 14.0 months, with the area under curve being 0.70 (sensitivity, 50.0%; specificity, 83.7%).

**Table 1 jcm-11-06245-t001:** Patient characteristics according to unstable flaps.

	Overall	Flap Group (*n* = 45)	Non-Flap Group (*n* = 20)	*p* Value
Age, yr ^a^	52.2 ± 12.6	49.6 ± 12.8	57.7 ± 11.7	0.036
Male/Female, *n*	29/36	21/24	8/12	0.788
BMI, kg/m^2 a^	24.8 ± 6.1	24.4 ± 3.9	25.8 ± 9.6	0.394
Lower limb alignment, deg ^a, b^	2.7 ± 2.7	2.3 ± 2.8	3.6 ± 2.4	0.116
Trauma history, *n*	29	22	7	0.418
Mechanical symptom, *n*	50	34	16	0.761
Symptom duration, month ^a^	13.0 ± 5.8	9.3 ± 4.4	20.9 ± 11.3	0.017
VAS score ^a^	5.9 ± 2.4	6.0 ± 2.5	5.8 ± 2.4	0.793
Lysholm score ^a^	67.7 ± 17.6	65.9 ± 19.5	71.6 ± 11.9	0.284
High-grade cartilage wear ^c^				
Medial femoral condyle	8	2	6	0.008
Medial tibial plateau	5	1	4	0.028
Subchondral BML	7	2	5	0.025

BMI, body mass index; VAS, visual analogue scale; BML, bone marrow lesion. ^a^ Data are reported as mean ± SD unless otherwise indicated. ^b^ Positive values indicate varus alignment, whereas negative values indicate valgus alignment. ^c^ High-grade cartilage wear was deemed as that of ≥grade 3 based on the Yulish grading system.

**Table 2 jcm-11-06245-t002:** Logistic regression analyses regarding unstable flaps.

	*p* Value		Exp(β) Coefficient (95% CI)	
	Univariate	Multivariate	Univariate	Multivariate
Age	0.046	0.258	0.94 (0.89–0.99)	0.96 (0.90–1.03)
Sex	0.618		1.31 (0.45–3.82)	
BMI	0.409		0.97 (0.89–1.05)	
Lower limb alignment	0.119		0.83 (0.65–1.05)	
Trauma history	0.301		1.78 (0.60–5.28)	
Mechanical symptom	0.695		0.77 (0.21–2.81)	
Symptom duration	0.028	0.026	0.99 (0.98–0.99)	0.99 (0.98–0.99)
VAS score	0.788		1.04 (0.81–1.33)	
Lysholm score	0.281		0.98 (0.94–1.02)	
High-grade cartilage wear ^a^				
Medial femoral condyle	0.011	0.017	0.11 (0.02–0.60)	0.06 (0.01–0.61)
Medial tibial plateau	0.038	0.95	0.09 (0.01–0.88)	1.00 (0.99–1.00)
Subchondral BML	0.027	0.187	0.14 (0.02–0.80)	0.25 (0.03–1.98)

BMI, body mass index; VAS, visual analogue scale; BML, bone marrow lesion. ^a^ High-grade cartilage wear was deemed as that of ≥grade 3 based on the Yulish grading system.

## Data Availability

Not applicable.

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
