# Peer review of "Hidden Unstable Flap Should Be Suspected in Treating Intractable Pain from Medial Meniscus Horizontal Tear"

_jcm, 2022, doi:10.3390/jcm11216245_

Round 1

Reviewer 1 Report

The study aims to find possible factors that can help identify the presence of hidden flap in patients with intractable pain from medial meniscus horizontal tear, who required arthroscopic treatment.

Detecting patients with a low probability of having a hidden flap could help to persevere with conservative treatments, avoiding perhaps unnecessary arthroscopy. Conversely, identifying patients with a hidden flap could simplify treatment by referring them more quickly to surgery.

I think the aim of the study is very interesting and potentially useful. The study is methodologically well conducted. The article is well written in all its parts (complete introduction, well-described methods, clear results, conclusions consistent with the results).

In fact, I think the presence of figures may help the understanding of the problem for readers less familiar with the topic. Therefore I would ask the authors to include a drawing of the meniscal lesion types and one or two intraoperative images of the flaps.

Thank you.

Author Response

As suggested, the figure of meniscal lesion has been added (Figure 1). 

Reviewer 2 Report

Dear Authors,

It is well designed and written. However, it should be noted that currently all inside repair of horizontal tears becomes more and more common, with large amount of studies supporting this way of treatment (i.e. doi: 10.1007/s00167-022-07133-w; doi: 10.1016/j.asmr.2021.01.018). Please discuss this issue in the introduction and maybe mention it in the discussion.

Author Response

As suggested, the descriptions regarding arthroscopic repair have been added in the introduction (Line 82-84) and the discussion (Line 188-189).